# Geometric Algebra Jordan–Wigner Transformation for Quantum Simulation

**DOI:** 10.3390/e26050410

**Published:** 2024-05-08

**Authors:** Grégoire Veyrac, Zeno Toffano

**Affiliations:** Laboratoire Signaux et Systèmes (L2S), UMR 8506, CentraleSupélec, Université Paris-Saclay, CNRS, 91190 Gif-sur-Yvette, France; gregoire.veyrac@student-cs.fr

**Keywords:** Geometric algebra, quantum computing, quantum simulation

## Abstract

Quantum simulation qubit models of electronic Hamiltonians rely on specific transformations in order to take into account the fermionic permutation properties of electrons. These transformations (principally the Jordan–Wigner transformation (JWT) and the Bravyi–Kitaev transformation) correspond in a quantum circuit to the introduction of a supplementary circuit level. In order to include the fermionic properties in a more straightforward way in quantum computations, we propose to use methods issued from Geometric Algebra (GA), which, due to its commutation properties, are well adapted for fermionic systems. First, we apply the Witt basis method in GA to reformulate the JWT in this framework and use this formulation to express various quantum gates. We then rewrite the general one and two-electron Hamiltonian and use it for building a quantum simulation circuit for the Hydrogen molecule. Finally, the quantum Ising Hamiltonian, widely used in quantum simulation, is reformulated in this framework.

## 1. Introduction

Analysis of large molecules is a great challenge in computational chemistry. Studying the Shrödinger equation in most cases does not lead to analytic solutions, and therefore requires one to rely on simulations. The major goal in computational chemistry is to find the state energy spectrum of many body interacting fermionic systems and to investigate its wavefunctions and structural properties. Classical chemical simulations permit one to tackle efficiently physical, chemical, and structural properties of molecules. Due to the increase in computational power over the past decades, direct simulations have been made possible; for example, Density Functional Theory and Molecular Dynamics methods have enabled calculations for systems of more than a thousand atoms [1,2], but these methods suffer from an exponentially growing computational time. Some alternative methods have been developed in order to ease the computations for complex molecules, such as quantum Monte Carlo [3], tensor networks [4], post Hartree Fock methods [5], Green’s function methods [6], many-body perturbation theory [7], etc. However, each method suffers from its own limitations; for example, up to now, tensor networks have been demonstrated to be efficient only for low entangled systems [8].

The general difficulty in simulating quantum mechanics with classical computers is linked to the exponential scaling of the Hilbert space dimension, with respect to the number of atoms in the molecule. In the 1980s, Richard Feynman [9] made the proposition to simulate quantum mechanics with a device capable of imitating the quantum rules rather than simulate it on a classical computer. This idea was taken further by David Deutsch and led to the concept of quantum computers [10]. Quantum computers and quantum simulation are today a major research subject, involving private companies such as IBM [11], Google [12], Quantinuum [13], and Pasqal [14], along with universities all around the world. The seeked goal is to provide an advantage over classical computers [15] and to create an efficient tool for simulating quantum physical systems [16].

In quantum simulation, the molecular Hamiltonian under the Born–Oppenheimer approximation (BOA), where the atomic nuclei are considered fixed, is represented in the second-quantized form [17], using fermionic annihilation and creation operators. These operators anticommute for distinct electrons in a molecule; this means that they do not behave as qubits in a quantum circuit . Quantum circuits use the operations of matrix product to apply consecutive gates and Kronecker products to compose different qubits. This leads to the usual quantum circuit diagram where the consecutive boxes (from left to right) represent the applied quantum gates and the horizontal parallel wires represent the composed qubits behaving as commuting bosons. This fact makes the translation from the Hamiltonian form to quantum gates not straightforward. To practically obtain a quantum circuit from a Hamiltonian, one uses specific transformations such as the Jordan–Wigner transformation (JWT) [17,18] or the Bravyi–Kitaev transformation [19,20], which reestablishes the correct commutation properties. Then the Suzuki–Trotter approximation [17] is used to build the entire equivalent circuit. It is then possible to extract the useful information about the Hamiltonian, ground energies, for example, using different quantum algorithms.

Although JWT is effective in going from one formalism to another, it is commonly applied in an heuristic ad-hoc way. One could wonder whether alternative mathematical formalisms could make the application of this transformation more natural. We will show here that Geometric Algebra (GA) is an interesting candidate.

GA combines the algebra of quaternions, discovered by W.H. Hamilton, permitting one to describe geometric operations such as rotations and Grassman algebra based on the notion of exterior product, used to create objects of higher dimensions such as areas and volumes. The synthesis and generalization was done by Clifford [21] by combining quaternions and Grassmann’s algebra into a single mathematical structure. Clifford himself named this structure “geometric algebra”, although the term Clifford Algebra (CA) is more often used in the literature.

It must be emphasized that for historical reasons most physicists did not adopt GA for calculations on vector spaces, preferring Gibbs vector analysis, which almost completely replaced quaternion algebra by the end of the 19th century. However, in the second half of the 20th century, some physicists, lead by David Hestenes [22], used GA methods in classical mechanics, electromagnetism, quantum physics, and special and general relativity [23].

GA now also has many other applications in applied sciences outside physics, such as robotics [24] and computer graphics [21]. More generally, GA has shown its potential for optimizing geometry-related problems [24]. Current research has also used GA in quantum computing [25,26,27,28]. Recently, a formalism for quantum computing was proposed built upon complex Clifford algebras using the Witt Basis (WB) [29].

In this paper, we provide a direct link between geometric algebra and quantum simulation, using both the WB and JWT. The goal here is to establish the equivalence between the algebra generated by the WB and the algebra generated by the creation and annihilation operators in the electronic Hamiltonians used in quantum simulation.

Section 2 recalls the basics of GA and then defines the concept of WB and of complex CA; it concludes with the spinor representation and the qubit analogy. The JWT is discussed in Section 3, and then the formulation of JWT in the framework of GA and WB is developed; the method is then applied to give the expressions of the usual 1-qubit and 2-qubit quantum gates and discusses the application on qubits in quantum circuits. Quantum simulation Hamiltonians in the BOA form are then expressed in the GA framework and a concrete example, the hydrogen molecule, is investigated in Section 4. In Section 5, an example of a general anisotropic XY Hamiltonian, which can be reduced to a quantum Ising Hamiltonian, commonly used as an ersatz in quantum simulation, is discussed in our framework.

## 2. Geometric Algebra (GA)

### 2.1. Clifford Algebra (CA)

A CA is defined using a real vector space, *V*, with a quadratic form, ϕ, over a field [30]. The algebra is produced by the quotient space $T\left(V\right)/I_{\phi }\left(V\right)$, where *T* is the tensor product of vector spaces and $I_{\phi }=v⊗v-\phi \left(v\right)$ is the ideal subset closed under multiplication.

For an orthonormal basis, $\left\{e_{1},\cdots ,e_{n}\right\}$, spanning the vector space, *V*, the algebra is generated by the basis verifying
(1)$$\begin{array}{c} e_{i}e_{j}+e_{j}e_{i}=2\phi \left(e_{i},e_{j}\right). \end{array}$$

An intuitive definition of a CA can be given using a finite dimensional vector space, *V*, equipped with the “geometric product” defined as the sum of an inner product and an outer product, which both have a geometrical meaning [21,22], as shown below.

#### 2.1.1. Outer Product and Blades

**Definition** **1.**
*Outer product*

*The outer product, noted by ∧, is an antisymmetric bilinear form giving the anticommutativity for the basis vectors:*

(2)
$$\begin{array}{c} e_{i}\wedge e_{j}=-e_{j}\wedge e_{i}. \end{array}$$



The outer product of two vectors forms a new object, called a bivector.

**Definition** **2.**
*Blade*

*The blades are generated by the outer product of vectors of V. The grade of the blade is the number of vectors composing the blade:*

$$\begin{array}{c} e_{i_{1}}\wedge \cdots \wedge e_{i_{r}} \end{array}$$

*is called an r-blade.*


By definition, 0-blades are the scalars.

1-blades correspond to the basis vectors.

2-blades (also called bivectors) can be interpreted as oriented surfaces (as shown on Figure 1), 3-blades (or trivectors) as oriented volumes, etc…

#### 2.1.2. Inner Product

**Definition** **3.**
*Inner product*

*The inner product is a bilinear symmetric non-degenerate form defined on V.*

*For an orthonormal basis, $\left\{e_{i}\right\}$, one has:*

(3)
$$\begin{array}{c} e_{i}.e_{j}=\left\{\begin{array}{cc} 1 & \;if\;i=j\in 〚1,p〛\\ -1 & \;if\;i=j\in 〚p+1,n〛\\ 0 & \;otherwise \end{array},\right. \end{array}$$

*where p≤n. This characterizes the CA, noted $C\;\ell_{p,q}$, with p+q=n. For a general signature, p denotes the numbers of positive norm basis vectors and q the number of negative norm basis vectors.*

*The definition of the inner product is extended to all blades by the formulae*

(4)
$$\begin{array}{c} (u.\left(a_{1}\wedge \cdots \wedge a_{r}\right))=\\ \sum_{k=1}^{p}{\left(-1\right)}^{k}\left(u.c_{k}\right)a_{1}\wedge \cdots a_{k-1}\wedge a_{k+1}\wedge \cdots \wedge a_{r} \end{array}$$


(5)
$$\begin{array}{c} \left(a_{1}\wedge \cdots \wedge a_{r}\right).\left(b_{1}\wedge \cdots \wedge b_{p}\right)=\\ \left(a_{1}\wedge \cdots \wedge a_{r-1}\right).\left(a_{r}.\left(b_{1}\wedge \cdots \wedge b_{p}\right)\right), \end{array}$$

*where $a_{1}\wedge \cdots \wedge a_{r}$ and $b_{1}\wedge \cdots \wedge b_{p}$ are r- and p-blades, and u is a vector. Then, the inner product lowers the grade of a blade.*


The subspace $Span\{e_{i}|1\leq i\leq p\}$ forms an euclidean space, and $Span\{e_{i}|p+1\leq i\leq n\}$ forms an anti-euclidean space with signature −1.

A CA $C\;\ell_{p,q}$ is thus composed by $p+q=dimC\;\ell_{p,q}$ possible unordered combinations of orthonormal basis vectors $\left\{e_{i}:i=1,2,...,n\right\}$, obeying (Equation 1).

#### 2.1.3. Geometric Product

**Definition** **4.**
*Geometric product*

*The geometric product is the sum of the outer and inner products:*

(6)
$$\begin{array}{c} xy=x.y+x\wedge y. \end{array}$$



In the case of a vector, the outer product is the antisymmetric part and the inner product is the symmetric part.

More generally, for an r-blade, $A_{r}$, and an p-blade, $B_{p}$, we have :(7)$$\begin{array}{c} A_{r}\wedge B_{p}={\left(-1\right)}^{rp}B_{p}\wedge A_{r} \end{array}$$(8)$$\begin{array}{c} A_{r}.B_{p}={\left(-1\right)}^{r(p+1)}B_{p}.A_{r}. \end{array}$$

Any combination made of non-linearly independent vectors vanishes because of the anticommutativity of the outer product. In this way, $C\;\ell_{p,q}$ can be represented as a direct sum of i-blade independent sub-spaces:(9)$$\begin{array}{c} C\;\ell_{p,q}=W_{0}⊕W_{1}⊕...⊕W_{i}⊕...⊕W_{p+q}. \end{array}$$

The highest blade element has rank p+q=n and is unique and squares to −1; it is named the *pseudoscalar*. The rank *n* of the pseudo-scalar is interesting as it specifies the dimension of the vector space over which the algebra is defined.

Moreover, the number of independent blades of rank k∈{0,…,n} is given by the binomial coefficient $\left(\;\;\begin{array}{c} n\\ k \end{array}\;\;\right)=\frac{n!}{k!(n-k)!}$. Meaning that the generic CA $C\;\ell_{p,q}$, with p+q=n, globally contains $2^{n}$ independent blades.

This last fact shows that one can look for a correspondence between the $2^{n}$ blades with a system of *n* Fermions (each Fermion possessing two degrees of freedom) or *n* qubits; this will be the object of the following paragraphs.

### 2.2. Mother Algebra and Witt Basis (WB)

The *mother algebra* is defined from the Euclidean vector space, *V*, and its dual $V^{*}$ [31]:(10)$$\begin{array}{c} R^{n,n}=V⊗V^{*}. \end{array}$$

The corresponding CA is noted as $R_{n,n}$, called the *mother algebra*. If $w_{i}$ is a basis of *V* and $w_{i}^{*}$ is a basis of $V^{*}$, then $R_{n,n}$ is the algebra generated by the blades:$$\begin{array}{c} w_{i_{1}}\ldots w_{i_{k}}w_{j_{1}}^{*}\ldots w_{j_{\ell }}^{*}, \end{array}$$
with 1≤k,ℓ≤n.

The inner product on $R_{n,n}$ is defined as follows:(11)$$\begin{array}{cc} w_{i}.w_{j} & =0 \end{array}$$(12)$$\begin{array}{cc} w_{i}^{*}.w_{j}^{*} & =0 \end{array}$$(13)$$\begin{array}{cc} w_{i}.w_{j}^{*} & =w_{j}^{*}.w_{i}=\frac{1}{2}\delta_{i,j}. \end{array}$$

Therefore, this set of vectors, $\left\{w_{i},w_{i}^{*}|1\leq i\leq n\right\}$, forms a non-orthonormal basis of the mother algebra. This basis is called the *Witt basis* (WB).

We can use the following reversion/conjugation properties to transform a blade into its dual when using the WB:

**Definition** **5.**
*Reversion/Hermitian conjugation*

*On a real Clifford space, the Hermitian conjugation is the anti-automorphism of $C_{p,q}$, which maps vectors to their dual:*


*$\forall v\in V,v^{†}=v^{*}$ and $v^{*†}=v$*


$\forall x,y\in C_{p,q},{\left(xy\right)}^{†}=y^{†}x^{†}$



$\forall x,y\in C_{p,q},a,b\in R,{\left(ax+by\right)}^{†}=ax^{†}+by^{†}$




The WB verifies the relations: (14)$$\begin{array}{cc} w_{j}w_{j}^{*}+w_{j}^{*}w_{j} & =1 \end{array}$$(15)$$\begin{array}{cc} w_{i}w_{j}+w_{j}w_{i} & =0 \end{array}$$(16)$$\begin{array}{cc} w_{i}^{*}w_{j}+w_{j}w_{i}^{*} & =0 \end{array}$$(17)$$\begin{array}{cc} w_{i}^{*}w_{j}^{*}+w_{j}^{*}w_{i}^{*} & =0 \end{array}$$(18)$$\begin{array}{cc} {\left(w_{j}^{*}\right)}^{2} & =0 \end{array}$$(19)$$\begin{array}{cc} {\left(w_{j}\right)}^{2} & =0. \end{array}$$

An orthonormal basis can be defined from the WB as:(20)$$\begin{array}{c} e_{i}=w_{i}+w_{i}^{*} \end{array}$$(21)$$\begin{array}{c} {\bar{\phi }}_{i}=w_{i}-w_{i}^{*}, \end{array}$$
giving $e_{i}^{2}=1$ and ${\bar{\phi }}_{i}^{2}=-1$.

Therefore, the mother algebra can be decomposed as a tensor product of two CA, one Euclidean, and the other anti-Euclidean:(22)$$\begin{array}{c} R_{n,n}=R_{n}⊗{\bar{R}}_{n}, \end{array}$$
where $R_{n}=C\;\ell \left(e_{1},\ldots ,e_{n}\right)$ and ${\bar{R}}_{n}=C\;\ell \left({\bar{\phi }}_{1},\ldots ,{\bar{\phi }}_{n}\right)$. Here, the $e_{i}$ are Euclidean and the ${\bar{\phi }}_{i}$ are anti-Euclidean vectors.

### 2.3. Witt Basis as a Complex CA

The previous properties can be interpreted as a complex structure of the mother algebra [28,30]. Indeed, by defining the vectors $\phi_{j}=i{\bar{\phi }}_{j}$, where *i* is the imaginary unit, we obtain the complex CA:(23)$$\begin{array}{c} R_{n,n}=R_{n}⊗iR_{n}^{\prime }, \end{array}$$
with $R_{n}^{\prime }=C\;\ell \left(\phi_{1},\ldots ,\phi_{n}\right)$, where now the vectors $\phi_{i}$ are Euclidean.

In this case, the WB can be written as:(24)$$\begin{array}{c} w_{i}=\frac{1}{2}\left(e_{i}+i\phi_{i}\right) \end{array}$$(25)$$\begin{array}{c} w_{i}^{*}=\frac{1}{2}\left(e_{i}-i\phi_{i}\right). \end{array}$$

As noted in [31], the anticommutation properties (Equation 14) to (19) of the WB are analogous to those of the creation operator, $c^{†}$, and annihilation operator, *c*, used in quantum physics for fermions, used to write Hamiltonians in the second quantization form.

For this reason, it will be more convenient in the following to use the fermion notation:(26)$$\begin{array}{c} w_{i}^{*}=c_{i} \end{array}$$(27)$$\begin{array}{c} w_{i}=c_{i}^{†}. \end{array}$$

### 2.4. Spinors for Quantum Computing

The WB can be used to build spinor spaces and also to represent qubit states.

**Definition** **6.**
*Spinor space*

*A spinor space is a minimal left ideal of the CA generated by a hermitian idempotent element. It can be written using the mother algebra $R_{n,n}$ of the WB [31]:*

(28)
$$\begin{array}{c} S_{n}=R_{n,n}I, \end{array}$$

*where $I\in R_{n,n}$, such that $I^{†}=I$ and is an idempotent $I=I^{2}$.*


The following commuting self-adjoint idempotent operators [32] are then defined:(29)$$\begin{array}{c} I_{j}=c_{j}c_{j}^{†} \end{array}$$(30)$$\begin{array}{c} K_{j}=c_{j}^{†}c_{j} \end{array}$$
for j∈〚1,n〛.

The property (Equation 14) translates into the closure relation for a single particle:(31)$$I_{j}+K_{j}=1_{j}.$$

Then, the spinor ideals can be expressed:(32)$$\begin{array}{c} S_{\left\{i_{1},\ldots ,i_{t}\right\}\left\{k_{1},\ldots ,k_{s}\right\}}=R_{n,n}I_{i_{1}}\ldots I_{i_{t}}K_{k_{1}}\ldots K_{k_{s}}, \end{array}$$
where $\left\{i_{1},\ldots ,i_{t}\right\}⊔\left\{k_{1},\ldots ,k_{s}\right\}=〚1,n〛$ are spinor spaces (the sets are disjoints).

The resolution of the identity is obtained for the entire vector space by using the complete set of idempotent operators:(33)$$\begin{array}{cc} 1 & =\prod_{i=1}^{n}\left(I_{i}+K_{i}\right)\\ \; & =\sum_{\left\{i_{1},\ldots ,i_{t}\right\}⊔\left\{k_{1},\ldots ,k_{s}\right\}=〚1,n〛}I_{i_{1}}\ldots I_{i_{t}}K_{k_{1}}\ldots K_{k_{s}}. \end{array}$$

This means that the entire algebra can be decomposed into a direct sum of $2^{n}$ spinor spaces.

Quantum computations with *n* qubits can then be described using these complex CA $R_{n,n}$ by means of the WB, as outlined in [29]; an n-qubit state vector becomes an element of a spinor.

As shown above in (34), the spinor space can be represented by one of the $2^{n}$ projectors $I_{i_{1}}\ldots I_{i_{t}}K_{k_{1}}\ldots K_{k_{s}}$. In quantum physics, the natural choice is the empty spinor space, where the projector $I=I_{1}\ldots I_{n}$ represents the vacuum state |0…0⟩.

Using the creation operators, $c_{i}^{†}$, and the annihilation operators, $c_{i}$, we have:(34)$$\begin{array}{cc} c_{i}I_{1}\ldots I_{n} & =0 \end{array}$$(35)$$\begin{array}{cc} c_{i}^{†}I_{1}\ldots I_{n} & =I_{1}\ldots I_{i-1}c_{i}^{†}I_{i+1}\ldots I_{n}. \end{array}$$

In this way, the n-qubit state can be represented in this spinor formulation :(36)$$\begin{array}{c} |x_{1}\ldots x_{n}\rangle ↔{\left(c_{1}^{†}\right)}^{x_{1}}\left(\ldots \right){\left(c_{n}^{†}\right)}^{x_{n}}I. \end{array}$$

In a more general formulation [32], one could equivalently consider the projector $I_{i_{1}}\ldots I_{i_{t}}K_{k_{1}}\ldots K_{k_{s}}$ as the original spinor, corresponding now to a qubit state where $i_{\ell }$ qubits are in state |0⟩ and $k_{\ell }$ qubits are in state |1⟩. In this paper, we will adopt the representation of (Equation 36).

## 3. Reformulation of the Jordan–Wigner Transformation (JWT)

### 3.1. Second Quantization

Let us consider a system made of *N* fermions. The state of the system is described by the state vector
(37)$$\begin{array}{c} \left|\psi \rangle =\right|x_{1},\ldots ,x_{j},\ldots ,\;x_{n}\rangle , \end{array}$$
where $x_{j}$ counts as the number of particles in the state *j*. Each $x_{j}$ is either 0 or 1 due to the Pauli exclusion principle.

The annihilation and creation operators, $c_{j}$ and $c_{j}^{†}$, give the following relations when applied to ψ:(38)$$\begin{array}{cc} c_{j}|x_{1},\ldots ,x_{j},\ldots ,\;x_{n}\rangle & ={\left(-1\right)}^{\sum_{k=1}^{j-1}x_{k}}x_{j}|x_{1},\ldots ,1-x_{j},\ldots ,\;x_{n}\rangle \end{array}$$(39)$$\begin{array}{cc} c_{j}^{†}|x_{1},\ldots ,x_{j},\ldots ,\;x_{n}\rangle & ={\left(-1\right)}^{\sum_{k=1}^{j-1}x_{k}}\left(1-x_{j}\right)|x_{1},\ldots ,1-x_{j},\ldots ,\;x_{n}\rangle , \end{array}$$
and they correspond to removing or adding a particle in the state *j*. These operators are not hermitian, so they do not correspond to observable quantities. The number operator
(40)$$\begin{array}{c} n_{j}=c_{j}^{†}c_{j} \end{array}$$
is hermitian and returns the numbers 0 for no-particle and 1 if the particle is present in the state *j*.

The creation and annihilation operators verify the following properties:(41)$$\begin{array}{cc} c_{j}c_{j}|\psi \rangle & ={\left(-1\right)}^{\sum_{k=1}^{j-1}x_{k}}x_{j}c_{j}|\psi_{x_{j}↔1-x_{j}}\rangle =x_{j}\left(1-x_{j}\right)|\psi \rangle =0 \end{array}$$(42)$$\begin{array}{cc} c_{j}^{†}c_{j}^{†}|\psi \rangle & ={\left(-1\right)}^{\sum_{k=1}^{j-1}x_{k}}x_{j}c_{j}^{†}|\psi_{x_{j}↔1-x_{j}}\rangle =x_{j}\left(1-x_{j}\right)|\psi \rangle =0 \end{array}$$(43)$$\begin{array}{cc} c_{j}c_{j}^{†}|\psi \rangle & ={\left(-1\right)}^{\sum_{k=1}^{j-1}x_{k}}1-x_{j}c_{j}|\psi_{x_{j}↔1-x_{j}}\rangle =\left(1-x_{j}\right)|\psi \rangle . \end{array}$$

The two first equations give zero because of the Pauli exclusion principle (i.e., $x_{j}\in \left\{0,1\right\}$). The last equation also signifies the closure that
(44)$$\begin{array}{c} \left\{c_{j},c_{j}^{†}\right\}=c_{j}c_{j}^{†}+c_{j}^{†}c_{j}=1. \end{array}$$

Moreover, the annihilation/creation operators acting on different states always anticommute ∀i≠j,
(45)$$\begin{array}{c} \left\{c_{i},c_{j}\right\}=\left\{c_{i}^{†},c_{j}\right\}=\left\{c_{i}^{†},c_{j}^{†}\right\}=0. \end{array}$$

It is clear that one again finds the same commutation properties as in the case of the WB discussed above in the framework of GA, but here we started exclusively from the quantum theory; this analogy between the two methods justifies our approach.

This representation is called *second quantization*; it is useful for modeling the Hamiltonians of molecules, where the states represent the possible orbital states for electrons and their spins. This representation can then be used to simulate the molecule by translating this Hamiltonian into quantum gates and applying a quantum algorithm to extract the orbital energies.

Creation and annihilation operators act on fermion states, which is why they anticommute (see (Equation 45)), while quantum gates act on qubits, which are represented by wires in a quantum circuit and behave as bosons whose creation and annihilation operators, $a_{i}^{†}$ and $a_{j}$, commute. Therefore, fermion creation and annihilation operators and qubit gates do not correspond to the same mathematical structure; this is the reason why a transformation is required to pass from one framework to the other [17].

### 3.2. Jordan–Wigner Transformation (JWT)

The Jordan–Wigner transformation (JWT) uses the following operators [17,33]: (46)$$\begin{array}{cc} Z_{j} & =1-2c_{j}^{†}c_{j} \end{array}$$(47)$$\begin{array}{cc} \sigma_{j}^{+} & =\prod_{k=1}^{j-1}\left(1-2c_{k}^{†}c_{k}\right)c_{j} \end{array}$$(48)$$\begin{array}{cc} \sigma_{j}^{-} & =\prod_{k=1}^{j-1}\left(1-2c_{k}^{†}c_{k}\right)c_{j}^{†}, \end{array}$$
where the $\sigma_{j}^{+}$ and $\sigma_{j}^{-}$ are spin operators related to the 1-qubit Pauli gates $X_{j}$ and $Y_{j}$ by
(49)$$\begin{array}{c} \sigma_{j}^{\pm }=\frac{X_{j}\pm iY_{j}}{2}. \end{array}$$
The inverse transformation is given by:(50)$$\begin{array}{c} c_{j}^{†}=\left(\prod_{k=1}^{j-1}Z_{k}\right)\sigma_{j}^{-} \end{array}$$(51)$$\begin{array}{c} c_{j}=\left(\prod_{k=1}^{j-1}Z_{k}\right)\sigma_{j}^{+} \end{array}$$

One verifies that for even products of these operators one finds again the usual 1-qubit projection operators:(52)$$\begin{array}{c} c_{j}^{†}c_{j}=\frac{1-Z_{j}}{2} \end{array}$$(53)$$\begin{array}{c} c_{j}c_{j}^{†}=\frac{1+Z_{j}}{2}. \end{array}$$

The inverse Jordan–Wigner transformation therefore enables one to translate the Hamiltonian of the second quantized form, written with the $c_{i}^{†}$ and $c_{i}$ operators into a quantum circuit.

### 3.3. Permutation Properties of the JWT Operators

When using the WB formalism for the elementary 1-qubit operators $c_{j},c_{j}^{†},c_{j}c_{j}^{†},c_{j}^{†}c_{j}$, the permutation of the operators and the qubits states is ruled by the following property:

**Proposition** **1.**
*For $\lambda_{j}\in \left\{c_{j},c_{j}^{†},c_{j}c_{j}^{†},c_{j}^{†}c_{j}\right\}$,*

(54)
$$\begin{array}{c} \lambda_{1}|x_{1}\rangle \ldots \lambda_{n}|x_{n}{\rangle =\left(-1\right)}^{\sum_{j=1}^{n}\left|\lambda_{j}\right|\sum_{k=1}^{j-1}x_{k}}\lambda_{1}\ldots \lambda_{n}|x_{1}\rangle \ldots |x_{n}\rangle , \end{array}$$

*where $|\lambda_{j}|$ is the number of Witt basis vectors contained in $\lambda_{j}$, (i.e., $|c_{j}|=1$, $|c_{j}c_{i}^{†}|=2$, …).*


**Proof.** The aim here is to exchange the lambdas and the kets thanks to the anticommutation properties of the WB (Equation 44). Then, we only have to prove that
(55)$$\begin{array}{c} \lambda_{j-1}|x_{j-1}\rangle \lambda_{j}={\left(-1\right)}^{|\lambda_{j}|x_{j-1}}\lambda_{j-1}\lambda_{j}|x_{j-1}\rangle . \end{array}$$Depending of the values of $\lambda_{j-1}$ and $|x_{j-1}\rangle$, the product $\lambda_{j-1}|x_{j-1}\rangle$ takes the values presented in Table 1.The WB vectors with different indices always anticommute; this means that, in the case of $\lambda_{j}\in \left\{c_{j},c_{j}^{†}\right\}$, $\lambda_{j}$ and $|x_{j-1}\rangle$ anticommute if $x_{j-1}=1$ and commute if $x_{j-1}=0$. In the case of $\lambda_{j}\in \left\{c_{j}^{†}c_{j},c_{j}c_{j}^{†}\right\}$, $\lambda_{j}$ always commutes with $|x_{j-1}\rangle$. This can be written as (Equation 55).The formula (Equation 54) can then be found by exchanging the $\lambda_{j}$ successively with all the previous $|x_{k}\rangle$.Let us recall that the operator $n_{j}$ given in (Equation 40) acts on the qubit state as
$$\begin{array}{c} n_{j}|x_{1}\ldots x_{n}\rangle =x_{j}|x_{1}\ldots x_{n}\rangle , \end{array}$$
which gives
$$\begin{array}{c} {\left(-1\right)}^{\sum_{j=1}^{n}\left|\lambda_{j}\right|\sum_{k=1}^{j-1}n_{k}}|x_{1}\ldots x_{n}\rangle ={\left(-1\right)}^{\sum_{j=1}^{n}\left|\lambda_{j}\right|\sum_{k=1}^{j-1}x_{k}}|x_{1}\ldots x_{n}\rangle . \end{array}$$The above operator can be written using the property ${\left(-1\right)}^{\sum_{k=1}^{j-1}n_{k}}=\prod_{j=1}^{n}\left(1-2c_{j}^{†}c_{j}\right)$.Then, using (Equation 54), we finally get
(56)$$\begin{array}{cc} \lambda_{1}|x_{1}\ldots \lambda_{n}|x_{n}\rangle & =\prod_{\begin{array}{c} j=1\\ \lambda_{j}\in \left\{c_{j},c_{j}^{†}\right\} \end{array}}^{n}\prod_{k=1}^{j-1}\left(1-2c_{k}^{†}c_{k}\right)\lambda_{1}\ldots \lambda_{n}|x_{1}\ldots x_{n}\rangle . \end{array}$$□

### 3.4. Reformulation of Quantum Gates

This result enables one to directly express quantum gates in the GA formalism using the JWT. The problem of sign change due to anticommutativity, which was thoroughly discussed in [29] for the translation of the tensor product of qubits in a GA framework, is here directly encoded inside the JWT.

The JWT applied to the basis matrices leads to: (57)$$\begin{array}{cc} |0\rangle {\left\langle 0\right|}_{j} & =c_{j}c_{j}^{†} \end{array}$$(58)$$\begin{array}{cc} |1\rangle {\left\langle 1\right|}_{j} & =c_{j}^{†}c_{j} \end{array}$$(59)$$\begin{array}{cc} |0\rangle {\left\langle 1\right|}_{j} & =\left(\prod_{k=1}^{j-1}c_{k}c_{k}^{†}-c_{k}^{†}c_{k}\right)c_{j} \end{array}$$(60)$$\begin{array}{cc} |1\rangle {\left\langle 0\right|}_{j} & =\left(\prod_{k=1}^{j-1}c_{k}c_{k}^{†}-c_{k}^{†}c_{k}\right)c_{j}^{†}, \end{array}$$
which can be easily inverted: (61)$$\begin{array}{cc} c_{j} & =\left(\prod_{k=1}^{j-1}Z_{k}\right)|0\rangle {\left\langle 1\right|}_{j} \end{array}$$(62)$$\begin{array}{cc} c_{j}^{†} & =\left(\prod_{k=1}^{j-1}Z_{k}\right)|1\rangle {\left\langle 0\right|}_{j}. \end{array}$$

**Theorem** **1.**
*JWT for 1-qubit gates*

*The Jordan–Wigner transformation using the Witt basis provides the following expressions for the Pauli 1-qubit gates:*

(63)
$$\begin{array}{cc} X_{j} & =|0\rangle {\left\langle 1\right|}_{j}+|1\rangle {\left\langle 0\right|}_{j}=\prod_{k=1}^{j-1}\left(1-2c_{j}^{†}c_{j}\right)\left(c_{j}^{†}+c_{j}\right) \end{array}$$


(64)
$$\begin{array}{cc} Y_{j} & =-i|0\rangle {\left\langle 1\right|}_{j}+i|1\rangle {\left\langle 0\right|}_{j}=\prod_{k=1}^{j-1}\left(1-2c_{j}^{†}c_{j}\right)\left(ic_{j}^{†}-ic_{j}\right) \end{array}$$


(65)
$$\begin{array}{cc} Z_{j} & =|0\rangle {\left\langle 0\right|}_{j}-|1\rangle {\left\langle 1\right|}_{j}=1-2c_{j}^{†}c_{j} \end{array}$$


*This means that, if $\forall j,\Lambda_{j}\in \left\{X_{j},Y_{j},Z_{j}\right\}$, then we have*

(66)
$$\begin{array}{c} \Lambda_{1}|x_{1}\rangle \ldots \Lambda_{n}|x_{n}\rangle =\Lambda_{1}\ldots \Lambda_{n}|x_{1}\rangle \ldots |x_{n}\rangle . \end{array}$$



### 3.5. Two-Qubit Systems

Let us consider the case of a 2-qubit system; this means that we work in the CA $R_{2,2}$. The WB elements reduce to $\left(w_{1}^{*},w_{1},w_{2}^{*},w_{2}\right)$, here noted as $\left(c_{1},c_{1}^{†},c_{2},c_{2}^{†}\right)$.

According to the preceding spinor formulation, the qubit states are represented by the following multivectors: (67)$$\begin{array}{c} |0\rangle ⊗|0\rangle =|00\rangle ↔I_{1}I_{2}=c_{1}c_{1}^{†}c_{2}c_{2}^{†} \end{array}$$(68)$$\begin{array}{c} |0\rangle ⊗|1\rangle =|00\rangle ↔c_{2}^{†}I_{1}I_{2}=c_{1}c_{1}^{†}c_{2}^{†} \end{array}$$(69)$$\begin{array}{c} |1\rangle ⊗|0\rangle =|10\rangle ↔c_{1}^{†}I_{1}I_{2}=c_{1}^{†}c_{2}c_{2}^{†} \end{array}$$(70)$$\begin{array}{c} |1\rangle ⊗|1\rangle =|11\rangle ↔c_{1}^{†}c_{2}^{†}I_{1}I_{2}=c_{1}^{†}c_{2}^{†} \end{array}$$

This shows that for 2-qubit states, the Kronecker product is equivalent to the geometric product in $R_{2,2}$. In order to introduce the 1-qubit gates in a 2-qubit circuit, we use the JWT in $R_{2,2}$. For the gates acting on the first qubit, the JWT simply returns:(71)$$\begin{array}{cc} X⊗1 & =c_{1}^{†}+c_{1} \end{array}$$(72)$$\begin{array}{cc} Y⊗1 & =ic_{1}^{†}-ic_{1} \end{array}$$(73)$$\begin{array}{cc} Z⊗1 & =c_{1}c_{1}^{†}-c_{1}^{†}c_{1} \end{array}$$

For the gates acting on the second qubit, however, the JWT adds a correction factor, $c_{1}c_{1}^{†}-c_{1}^{†}c_{1}$, to the *X* and *Y* gates:(74)$$\begin{array}{cc} 1⊗X & =\left(c_{1}c_{1}^{†}-c_{1}^{†}c_{1}\right)\left(c_{2}^{†}+c_{2}\right) \end{array}$$(75)$$\begin{array}{cc} 1⊗Y & =\left(c_{1}c_{1}^{†}-c_{1}^{†}c_{1}\right)\left(ic_{2}^{†}-ic_{2}\right) \end{array}$$(76)$$\begin{array}{cc} 1⊗Z & =\left(c_{1}c_{1}^{†}-c_{1}^{†}c_{1}\right)\left(c_{2}c_{2}^{†}-c_{2}^{†}c_{2}\right). \end{array}$$

General 2-qubit gates can be written using the sums and products of the above 1-qubit gates. For instance, for the controlled-NOT, controlled-Z, and SWAP gates, we have, respectively:(77)$$\begin{array}{cc} CNOT & =\frac{1}{2}\left(1⊗1+1⊗X+Z⊗1-Z⊗X\right) \end{array}$$(78)$$\begin{array}{cc} CZ & =\frac{1}{2}\left(1⊗1+1⊗Z+Z⊗1-Z⊗Z\right) \end{array}$$(79)$$\begin{array}{cc} SWAP & =\frac{1}{2}\left(1⊗1+X⊗X+Y⊗Y+Z⊗Z\right), \end{array}$$
giving in our formulation in $R_{2,2}$:$$\begin{array}{cc} CNOT & =c_{1}c_{1}^{†}-c_{1}^{†}c_{1}\left(c_{2}^{†}+c_{2}\right)\\ CZ & =c_{1}c_{1}^{†}+c_{1}^{†}c_{1}\left(c_{2}c_{2}^{†}-c_{2}^{†}c_{2}\right)\\ SWAP & =c_{1}c_{1}^{†}c_{2}c_{2}^{†}+c_{1}^{†}c_{1}c_{2}^{†}c_{2}+c_{1}^{†}c_{2}-c_{1}c_{2}^{†}. \end{array}$$

### 3.6. Controlled Gates

In particular, one can derive the expression of controlled gates in $R_{n,n}$. Let $A_{k}$ be an arbitrary 1-qubit quantum gate acting on the qubit *k*. The controlled gate $C{\left(A_{k}\right)}_{j}$ (where *j* is the control qubit and *k* the target) is given by: (80)$$\begin{array}{cc} C{\left(A_{k}\right)}_{j} & =|0\rangle \langle 0{|}_{j}+|1\rangle \langle 1{|}_{j}A_{k} \end{array}$$(81)$$\begin{array}{cc} \; & =c_{j}c_{j}^{†}+c_{j}^{†}c_{j}A_{k} \end{array}$$

## 4. Application to Quantum Simulation

### 4.1. Electronic Hamiltonians for Quantum Simulation

The aim of quantum simulation is to analyze the electronic structure and the associated energies of a molecule. The Born–Oppenheimer approximation (BOA) is commonly used where all nuclei of the molecule are assumed to stay at fixed positions since nuclei have a mass larger than electrons by a factor of $10^{3}$. Therefore, the dynamics of the molecule is described by the electronic Hamiltonian, *H*, which is the sum of a single electron Hamiltonian, $H^{\left(1\right)}$, and a two-electron Hamiltonian, $H^{\left(2\right)}$. The wave function corresponding to the different energy levels, *E* (eigenvalues), of the molecule is an eigenfunction |ψ⟩ of the operator *H* given by the Shrödinger equation: (82)$$\begin{array}{c} H|\psi \rangle =E|\psi \rangle \end{array}$$
The ground state, $\left|\psi_{g}\right\rangle$, corresponds to the lowest energy, $E_{g}$. To describe electrons in the second quantization, we use a basis of *N* orbital states, noted as $\left|\chi_{j}\right\rangle$. Each wave function, $\chi_{j}$, is a single electron wave function and is the tensor product of a spatial component, |ϕ⟩, and a spin component, |↑⟩ or |↓⟩.

In the second quantization formalism, the single-electron and two-electron parts of the BOA Hamiltonians are based only on the relative electron dynamics and are expressed as:(83)$$\begin{array}{c} H^{\left(1\right)}=\sum_{i,j}h_{ij}c_{i}^{†}c_{j} \end{array}$$(84)$$\begin{array}{c} H^{\left(2\right)}=\sum_{i,j,k,\ell }h_{ijk\ell }c_{i}^{†}c_{j}^{†}c_{k}c_{\ell }. \end{array}$$

The first coefficient, $h_{ij}$, is the single-electron overlap integral; it consists of the kinetic energy and the potential energy for the one electron interaction from state *i* to state *j*. The coefficients $h_{ii}$ correspond to the unperturbed self-energy of the state, $\left|\chi_{i}\right\rangle$.

The second coefficient $h_{ijkl}$ is the two-electron overlap integral which corresponds to the different interaction energies of a system of 2 electrons initially in states *i* and *j* ending up in states *k* and *ℓ*. The two-electron interaction comprises : the Coulomb (i=ℓ, j=k) and exchange (i=k, j=ℓ) interactions which represent respectively the interactions between electrons and between electrons and nuclei, the double excitation operator (i≠j≠k≠ℓ) corresponds to the excitation of both electrons in all four orbital exchanges.
(85)$$\begin{array}{cc} h_{ij} & =\langle \chi_{i}|H^{\left(1\right)}|\chi_{j}\rangle \end{array}$$
(86)$$\begin{array}{cc} \; & =\int_{R}\chi_{i}^{*}\left(x\right)\left(-\frac{\hbar^{2}}{2m_{e}}\nabla^{2}-\sum_{\alpha =1}^{A}\frac{Z_{\alpha }e^{2}}{4\pi \epsilon_{0}∥x-x_{\alpha }∥}\right)\chi_{j}\left(x\right)dx \end{array}$$
(87)$$\begin{array}{cc} h_{ijk\ell } & =\langle \chi_{i}\chi_{j}|H^{\left(2\right)}|\chi_{k}\chi_{\ell }\rangle \end{array}$$
(88)$$\begin{array}{cc} \; & =\int_{R}\frac{\chi_{i}^{*}\left(x_{1}\right)\chi_{j}^{*}\left(x_{2}\right)\chi_{k}\left(x_{2}\right)\chi_{\ell }\left(x_{1}\right)e^{2}}{4\pi \epsilon_{0}∥x_{1}-x_{2}∥}dx_{1}dx_{2} \end{array}$$

The coefficients $h_{ij}$ and $h_{ijk\ell }$ for a given set of orbitals, $\left\{|\chi_{j}\rangle \right\}$, can be computed classically by solving the Hartree–Fock equations [17].

### 4.2. Hamiltonian Operators in the WB

In order to simulate quantum systems using the WB, we express the different contributions to the molecular Hamiltonians; these are shown in Table 2 [20].

In quantum simulations, we are also interested in the exponential of these operators. We will compute them in the following sections. The decomposition of an exponential form is based on the following quantum gates: (89)$$\begin{array}{c} T_{j}\left(\theta \right)=\left(\begin{array}{cc} 1 & 0\\ 0 & e^{-i\theta } \end{array}\right)=|0\rangle \langle 0|+e^{-i\theta }|1\rangle \langle 1|=c_{j}c_{j}^{†}+e^{-i\theta }c_{j}^{†}c_{j} \end{array}$$
and
(90)$$\begin{array}{c} R_{j}\left(\theta \right)=\left(\begin{array}{cc} e^{i\theta } & 0\\ 0 & e^{-i\theta } \end{array}\right)=e^{i\theta }|0\rangle \langle 0|+e^{-i\theta }|1\rangle \langle 1|=e^{i\theta }c_{j}c_{j}^{†}+e^{-i\theta }c_{j}^{†}c_{j}. \end{array}$$

The index corresponds to the position of the qubit affected by the operator.

#### 4.2.1. Energy Number Operator

The energy number operator, $H_{n_{i}}$, gives the electron energy of the *i*-th state. It is the product of the number operator $c_{i}^{†}c_{j}$ by $h_{ii}$, which is the mean value of *H* with respect to $\left|\chi_{i}\right\rangle$, as defined in Section 4.1.
(91)$$\begin{array}{c} H_{n_{i}}=h_{ii}c_{i}^{†}c_{i} \end{array}$$

The number operator is idempotent:(92)$$\begin{array}{c} {\left(c_{i}^{†}c_{i}\right)}^{2}=c_{i}^{†}c_{i}. \end{array}$$

Therefore, we have the exponential: $$\begin{array}{cc} e^{-iH_{n_{i}}t} & =1+\left(e^{-ih_{ii}t}-1\right)c_{i}^{†}c_{i}\\ \; & =c_{i}^{†}c_{i}+c_{i}c_{i}^{†}+\left(e^{-ih_{ii}t}-1\right)c_{i}^{†}c_{i}\\ \; & =c_{i}c_{i}^{†}+e^{-ih_{ii}t}c_{i}^{†}c_{i}\\ \; & =|0\rangle \langle 0|+e^{-ih_{ii}t}|1\rangle \langle 1|\\ \; & =T_{i}\left(h_{ii}t\right). \end{array}$$
The corresponding circuit is given in Figure 2.

#### 4.2.2. Coulomb Operator

The Coulomb operators describe Coulomb interactions between two electrons in states *i* and *j*. They are written as:(93)$$\begin{array}{c} H_{c_{ij}}=h_{ijji}c_{i}^{†}c_{j}^{†}c_{j}c_{i}. \end{array}$$

The product of the two number operators $c_{i}^{†}c_{i}$ and $c_{j}^{†}c_{j}$ is still idempotent:(94)$$\begin{array}{c} {\left(c_{i}^{†}c_{j}^{†}c_{j}c_{i}\right)}^{2}=c_{i}^{†}c_{j}^{†}c_{j}c_{i}. \end{array}$$

It is then possible to decompose the operator into the sum of involutive operators to obtain
$$\begin{array}{cc} c_{i}^{†}c_{j}^{†}c_{j}c_{i} & =c_{i}^{†}c_{i}c_{j}^{†}c_{j}\\ \; & =\frac{1}{4}\left(1-Z_{i}\right)\left(1-Z_{j}\right), \end{array}$$
where $Z_{j}=c_{j}c_{j}^{†}-c_{j}^{†}c_{j}$. Then, we have the exponential:$$\begin{array}{cc} \; & e^{-iH_{c_{ij}}t}\\ & =e^{-ih_{ijji}\frac{1}{4}\left(1-Z_{i}\right)\left(1-Z_{j}\right)t}\\ \; & =e^{-\frac{h_{ijji}t}{4}}\left(\cos\left(\frac{h_{ijji}}{4}\right)+i\sin\left(\frac{h_{ijji}}{4}\right)Z_{i}\right)\left(\cos\left(\frac{h_{ijji}}{4}\right)+i\sin\left(\frac{h_{ijji}}{4}\right)Z_{j}\right)\left(\cos\left(\frac{h_{ijji}}{4}\right)-i\sin\left(\frac{h_{ijji}}{4}\right)Z_{i}Z_{j}\right) \end{array}$$
besides, we have
$$\begin{array}{cc} \cos\left(\frac{h_{ijji}t}{4}\right)+i\sin\left(\frac{h_{ijji}t}{4}\right)Z_{j} & =\cos\left(\frac{h_{ijji}t}{4}\right)\left(c_{j}c_{j}^{†}+c_{j}^{†}c_{j}\right)+i\sin\left(\frac{h_{ijji}t}{4}\right)\left(c_{j}c_{j}^{†}-c_{j}^{†}c_{j}\right)\\ \; & =e^{\frac{ih_{ijji}t}{4}}c_{j}c_{j}^{†}+e^{-\frac{ih_{ijji}t}{4}}c_{j}^{†}c_{j}\\ \; & =R_{j}\left(\frac{h_{ijji}t}{4}\right) \end{array}$$
and
$$\begin{array}{cc} \; & \cos\left(\frac{h_{ijji}t}{4}\right)-i\sin\left(\frac{h_{ijji}t}{4}\right)Z_{i}Z_{j}\\ & =\cos\left(\frac{h_{ijji}t}{4}\right)\left(c_{j}c_{j}^{†}+c_{j}^{†}c_{j}\right)\left(c_{i}c_{i}^{†}+c_{i}^{†}c_{i}\right)+i\sin\left(\frac{h_{ijji}t}{4}\right)\left(c_{j}c_{j}^{†}-c_{j}^{†}c_{j}\right)\left(c_{i}c_{i}^{†}-c_{i}^{†}c_{i}\right)\\ \; & =e^{\frac{ih_{ijji}t}{4}}\left(c_{j}c_{j}^{†}c_{i}c_{i}^{†}+c_{j}^{†}c_{j}c_{i}^{†}c_{i}\right)-e^{-\frac{ih_{ijji}t}{4}}\left(c_{j}^{†}c_{j}c_{i}c_{i}^{†}+c_{j}c_{j}^{†}c_{i}^{†}c_{i}\right)\\ \; & =\left(c_{i}c_{i}^{†}-c_{i}^{†}c_{i}\left(c_{j}^{†}+c_{j}\right)\right)\left(e^{\frac{ih_{ijji}t}{4}}c_{j}c_{j}^{†}+e^{-\frac{ih_{ijji}t}{4}}c_{j}^{†}c_{j}\right)\left(c_{i}c_{i}^{†}-c_{i}^{†}c_{i}\left(c_{j}^{†}+c_{j}\right)\right)\\ \; & =\left(CNOT_{ij}\right)\left(R_{j}\left(\frac{h_{ijji}t}{4}\right)\right)\left(CNOT_{ij}\right) \end{array}$$

Thus, the exponential of the Coulomb operator is
(95)$$\begin{array}{c} e^{-iH_{c_{ij}}t}=R_{i}\left(\frac{h_{ijji}t}{4}\right)R_{j}\left(\frac{h_{ijji}t}{4}\right)\left(CNOT_{ij}\right)\left(R_{j}\left(\frac{h_{ijji}t}{4}\right)\right)\left(CNOT_{ij}\right). \end{array}$$
The corresponding circuit is given in Figure 3.

#### 4.2.3. Excitation Operator

The excitation operator has the form:(96)$$\begin{array}{c} H_{ex_{ij}}=h_{ij}\left(c_{i}^{†}c_{j}+c_{j}^{†}c_{i}\right). \end{array}$$

Then, knowing that
(97)$$\begin{array}{cc} {\left(c_{i}^{†}c_{j}+c_{j}^{†}c_{i}\right)}^{2} & =c_{i}^{†}c_{i}c_{j}c_{j}^{†}+c_{i}c_{i}^{†}c_{j}^{†}c_{j} \end{array}$$
(98)$$\begin{array}{cc} {\left(c_{i}^{†}c_{j}+c_{j}^{†}c_{i}\right)}^{3} & =c_{i}^{†}c_{j}+c_{j}^{†}c_{i}, \end{array}$$
its exponential becomes
$$\begin{array}{cc} e^{-iH_{ex_{ij}}t} & =1+\left(\cos(h_{ij}t)-1\right)\left(c_{i}^{†}c_{i}c_{j}c_{j}^{†}+c_{i}c_{i}^{†}c_{j}^{†}c_{j}\right)-i\sin\left(h_{ij}t\right)\left(c_{i}^{†}c_{j}+c_{j}^{†}c_{i}\right). \end{array}$$

In order to express the Hamiltonians using known gates, we will use the JWT. For the sin part, assuming i<j, we obtain
$$\begin{array}{cc} c_{i}^{†}c_{j}+c_{j}^{†}c_{i} & =\left(\prod_{k=0}^{i-1}Z_{k}\right)\left(|1\rangle \langle 0{|}_{i}\right)\left(\prod_{k=0}^{j-1}Z_{k}\right)\left(|0\rangle \langle 1{|}_{j}\right)-\left(\prod_{k=0}^{i-1}Z_{k}\right)\left(|0\rangle \langle 1{|}_{i}\right)\left(\prod_{k=0}^{j-1}Z_{k}\right)\left(|1\rangle \langle 0{|}_{j}\right)\\ \; & =\left(\prod_{k=i+1}^{j-1}Z_{k}\right)\left(|10\rangle \langle 01{|}_{ij})+|01\rangle \langle 10{|}_{ij}\right), \end{array}$$
since $|1\rangle \langle 0{|}_{i}Z_{i}=|1\rangle \langle 0{|}_{i}$ and $|0\rangle \langle 1{|}_{i}Z_{i}=-|0\rangle \langle 1{|}_{i}$.

For the cos part, we have
$$\begin{array}{cc} c_{i}^{†}c_{i}c_{j}c_{j}^{†}+c_{i}c_{i}^{†}c_{j}^{†}c_{j} & =|10\rangle \langle 10{|}_{ij}+|01\rangle \langle 01{|}_{ij}, \end{array}$$
which leads to the final form
(99)$$\begin{array}{cc} e^{-iH_{ex_{ij}}t}= & 1+\left(\cos\left(h_{ij}t\right)-1\right)\left(|10\rangle \langle 10{|}_{ij}+|01\rangle \langle 01{|}_{ij}\right)\\ & -i\sin\left(h_{ij}t\right)\left(\prod_{k=i+1}^{j-1}Z_{k}\right)\left(|10\rangle \langle 01{|}_{ij}+|01\rangle \langle 10{|}_{ij}\right). \end{array}$$

If j=i+1 and $h_{ij}t=\frac{\pi }{2}$, this operator is the swap gate applied to the qubits *i* and *j*, and a phase factor on two states. More generally, it is related to the exponential of the swap gate, as shown on Figure 4. If j>i+1, the phase inside the exponential can be transformed into its opposite depending on the state of the intermediate qubits due to the *Z* gates in the sin term of (Equation 99).
(100)$$\begin{array}{cc} e^{-iH_{ex_{ij}}t} & =1+\left(\cos\left(h_{ij}t\right)-1\right)\left(|10\rangle \langle 10{|}_{ij}+|01\rangle \langle 01{|}_{ij}\right)-i\sin\left(h_{ij}t\right)\left(|10\rangle \langle 01{|}_{ij}+|01\rangle \langle 10{|}_{ij}\right)\\ & =\left[e^{-ih_{ij}t{SWAP}_{ij}}\right]\left[C{\left(T_{j}\left(-h_{ij}t\right)\right)}_{i}\right]\left[X_{i}X_{j}C{\left(T_{j}\left(-h_{ij}t\right)\right)}_{i}X_{i}X_{j}\right], \end{array}$$
where the controlled *T* gates come from the phase factor on the states $|10\rangle \langle 01{|}_{ij}$ and $|01\rangle \langle 10{|}_{ij}$.

#### 4.2.4. Number-Excitation Operators

The number-excitation operators are:(101)$$\begin{array}{c} H_{nex_{ijk}}=h_{ijjk}\left(c_{i}^{†}c_{j}^{†}c_{j}c_{k}+c_{k}^{†}c_{j}^{†}c_{j}c_{i}\right). \end{array}$$

They obey the same kind of relations as the excitation operator:(102)$$\begin{array}{cc} {\left(c_{i}^{†}c_{j}^{†}c_{j}c_{k}+c_{k}^{†}c_{j}^{†}c_{j}c_{i}\right)}^{2} & =c_{i}^{†}c_{i}c_{j}^{†}c_{j}c_{k}c_{k}^{†}+c_{i}c_{i}^{†}c_{j}^{†}c_{j}c_{k}^{†}c_{k} \end{array}$$(103)$$\begin{array}{cc} {\left(c_{i}^{†}c_{j}^{†}c_{j}c_{k}+c_{k}^{†}c_{j}^{†}c_{j}c_{i}\right)}^{3} & =c_{i}^{†}c_{j}^{†}c_{j}c_{k}+c_{k}^{†}c_{j}^{†}c_{j}c_{i}. \end{array}$$

Similarly, the exponential gives:$$\begin{array}{cc} e^{-iH_{nex_{ijk}}t}= & \;1+\left(\cos(h_{ij}t)-1\right)\left(c_{i}^{†}c_{i}c_{j}^{†}c_{j}c_{k}c_{k}^{†}+c_{i}c_{i}^{†}c_{j}^{†}c_{j}c_{k}^{†}c_{k}\right)\\ & -i\sin\left(h_{ij}t\right)\left(c_{i}^{†}c_{j}^{†}c_{j}c_{k}+c_{k}^{†}c_{j}^{†}c_{j}c_{i}\right)\\ = & \;1+c_{j}^{†}c_{j}\left(\left(\cos(h_{ij}t)-1\right)\left(c_{i}^{†}c_{i}c_{k}c_{k}^{†}+c_{i}c_{i}^{†}c_{k}^{†}c_{k}\right)-i\sin\left(h_{ij}t\right)\left(c_{i}^{†}c_{k}+c_{k}^{†}c_{i}\right)\right)\\ = & \;c_{j}c_{j}^{†}+c_{j}^{†}c_{j}\left(1+\left(\cos(h_{ij}t)-1\right)\left(c_{i}^{†}c_{i}c_{k}c_{k}^{†}+c_{i}c_{i}^{†}c_{k}^{†}c_{k}\right)-i\sin\left(h_{ij}t\right)\left(c_{i}^{†}c_{k}+c_{k}^{†}c_{i}\right)\right) \end{array}$$

In this way, we obtain the gate of the excitation operator controlled by *j*:(104)$$\begin{array}{c} e^{-ih_{ijjk}\left(c_{i}^{†}c_{j}^{†}c_{j}c_{k}+c_{k}^{†}c_{j}^{†}c_{j}c_{i}\right)t}=C{\left(e^{-iH_{ex_{ik}}t}\right)}_{j}. \end{array}$$

This last result is quite natural, since the number-excitation operator results from the combination of the number operator on qubit *j* and the excitation operator between qubits *i* and *k*. The number operator returns the number of qubits in the state *j*, which is either 0 or 1, so it is the same as controlling the excitation operator by qubit *j*. Figure 5 represents the corresponding circuit in the case where j=i+1.

#### 4.2.5. Double Excitation Operators

The double excitation operators are:(105)$$\begin{array}{c} H_{dex_{ijk\ell }}=h_{ijk\ell }\left(c_{i}^{†}c_{j}^{†}c_{k}c_{\ell }+c_{\ell }^{†}c_{k}^{†}c_{j}c_{i}\right). \end{array}$$

They obey the same kind of relations as the excitation operator:(106)$$\begin{array}{cc} {\left(c_{i}^{†}c_{j}^{†}c_{k}c_{\ell }+c_{\ell }^{†}c_{k}^{†}c_{j}c_{i}\right)}^{2} & =c_{i}^{†}c_{i}c_{j}^{†}c_{j}c_{k}c_{k}^{†}c_{\ell }c_{\ell }^{†}+c_{i}c_{i}^{†}c_{j}^{†}c_{j}c_{k}^{†}c_{k}c_{\ell }^{†}c_{\ell } \end{array}$$(107)$$\begin{array}{cc} {\left(c_{i}^{†}c_{j}^{†}c_{k}c_{\ell }+c_{\ell }^{†}c_{k}^{†}c_{j}c_{i}\right)}^{3} & =c_{i}^{†}c_{j}^{†}c_{k}c_{\ell }+c_{\ell }^{†}c_{k}^{†}c_{j}c_{i}. \end{array}$$

Similarly, assuming i<j<k<ℓ, the exponential gives:(108)$$\begin{array}{cc} e^{-iH_{dex_{ijk\ell }}t}= & 1+\left(\cos(h_{ijk\ell }t)-1\right)\left(c_{i}^{†}c_{i}c_{j}^{†}c_{j}c_{k}c_{k}^{†}c_{\ell }c_{\ell }^{†}+c_{i}c_{i}^{†}c_{j}^{†}c_{j}c_{k}^{†}c_{k}c_{\ell }^{†}c_{\ell }\right)\\ & -i\sin\left(h_{ijk\ell }t\right)\left(c_{i}^{†}c_{j}^{†}c_{k}c_{\ell }+c_{\ell }^{†}c_{k}^{†}c_{j}c_{i}\right)\\ = & 1+\left(\cos(h_{ijk\ell }t)-1\right)\left(|1100\rangle \langle 1100{|}_{ijk\ell }+|0011\rangle \langle 0011{|}_{ijk\ell }\right)\\ & -i\sin\left(h_{ijk\ell }t\right)\left(\prod_{p=i+1}^{j-1}Z_{p}\right)\left(\prod_{p=k+1}^{\ell -1}Z_{p}\right)\left(|0011\rangle \langle 1100{|}_{ijk\ell }+|1100\rangle \langle 0011{|}_{ijk\ell }\right). \end{array}$$

If the condition i<j<k<ℓ is not verified, the *Z* gates do not simplify in the same way, but the global form of the operator remains the same; if $h_{ijk\ell }t=\frac{\pi }{2}$, similarly to the case of the simple excitation operator, we obtain a kind of 16×16 swap gate. Here again, it is possible to write the general form of (Equation 108) as a power of that matrix, with phase corrections applied to some states (either the 2 exchanged states or the 14 others). This matrix exchanges two composite sates (not two qubits).

### 4.3. Hydrogen Molecule

All the preceding results can be used to express the Hamiltonian of a hydrogen molecule, $H_{2}$. In the BOA, the dynamics of the nuclei can be separated from the dynamics of the electrons. In this case, the wave function for each electron can be deduced from the system’s symmetries. If $\phi_{1}$ and $\phi_{2}$ are the spatial wave functions of one single hydrogen, then the system can be described through the symmetric and antisymmetric spatial wave functions $\phi_{+}=\phi_{1}+\phi_{2}$ and $\phi_{-}=\phi_{1}-\phi_{2}$ [17]. Taking spin into account, the four basis states are then: (109)$$\begin{array}{c} |\xi_{1}\rangle =|\phi_{+}\rangle |↑\rangle ;\;|\xi_{2}\rangle =|\phi_{+}\rangle |↓\rangle ;\;|\xi_{2}\rangle =|\phi_{-}\rangle |↑\rangle ;\;|\xi_{4}\rangle =|\phi_{-}\rangle |↓\rangle , \end{array}$$

The BOA Hamiltonian of the hydrogen molecule in the second quantization given in (Equation 83) and (84) can thus be written directly in the WB:(110)$$\begin{array}{cc} H & =H^{\left(1\right)}+H^{\left(2\right)}, \end{array}$$
where
(111)$$\begin{array}{cc} H^{\left(1\right)}= & h_{11}c_{1}^{†}c_{1}+h_{22}c_{2}^{†}c_{2}+h_{33}c_{3}^{†}c_{3}+h_{44}c_{4}^{†}c_{4} \end{array}$$
(112)$$\begin{array}{cc} H^{\left(2\right)}= & h_{1221}c_{1}^{†}c_{2}^{†}c_{2}c_{1}+h_{3443}c_{3}^{†}c_{4}^{†}c_{4}c_{3}+h_{1441}c_{1}^{†}c_{4}^{†}c_{4}c_{1}+h_{2332}c_{2}^{†}c_{3}^{†}c_{3}c_{2}\\ & +\left(h_{1331}-h_{1313}\right)c_{1}^{†}c_{3}^{†}c_{3}c_{1}+\left(h_{2442}-h_{2424}\right)c_{2}^{†}c_{4}^{†}c_{4}c_{2}\\ & +ℜ\left(h_{1423}\right)\left(c_{1}^{†}c_{4}^{†}c_{2}c_{3}+c_{3}^{†}c_{2}^{†}c_{4}c_{1}\right)+ℜ\left(h_{1243}\right)\left(c_{1}^{†}c_{2}^{†}c_{4}c_{3}+c_{3}^{†}c_{4}^{†}c_{2}c_{1}\right)\\ & +ℑ\left(h_{1423}\right)\left(c_{1}^{†}c_{4}^{†}c_{2}c_{3}+c_{3}^{†}c_{2}^{†}c_{4}c_{1}\right)+ℑ\left(h_{1243}\right)\left(c_{1}^{†}c_{2}^{†}c_{4}c_{3}+c_{3}^{†}c_{4}^{†}c_{2}c_{1}\right). \end{array}$$

$H^{\left(1\right)}$ is the Hamiltonian for isolated electrons (kinetic energy and electron–proton interaction) and therefore only contains number operators. $H^{\left(2\right)}$ is the Hamiltonian for electron–electron interactions, so it uses the Coulomb and double excitation operators in $H^{\left(2\right)}$. Using the Trotter–Suzuki approximation, the results from Section 4.2 lead to the unitary:(113)$$\begin{array}{cc} e^{-iHt}≃ & \left[\left(\prod_{j=1}^{4}T_{j}\left(h_{jj}\Delta t\right)\right)\left(\prod_{1\leq j<k\leq 4}R_{j}\left(\frac{\theta_{jk}\Delta t}{4}\right)R_{k}\left(\frac{\theta_{jk}\Delta t}{4}\right)\left(CNOT_{jk}\right)\left(R_{k}\left(\frac{\theta_{jk}\Delta t}{4}\right)\right)\left(CNOT_{jk}\right)\right)\right.\\ \; & {\left.\left(D_{1423}\left(ℜ\left(h_{1423}\right)t\right)D_{1243}\left(ℜ\left(h_{1243}\right)t\right)D_{1423}\left(ℑ\left(h_{1423}\right)t\right)D_{1243}\left(ℑ\left(h_{1243}\right)t\right)\right)\right]}^{\frac{t}{\Delta t}}, \end{array}$$
where $\theta_{jk}=\left\{\begin{array}{cc} h_{jkkj}\; & \mathrm{if}\;j+k\;\mathrm{odd}\\ h_{jkkj}-h_{jkjk}\; & \mathrm{if}\;j+k\;\mathrm{even} \end{array}\right.$ and $D_{ijk\ell }$ is the exponential of the double excitation operator, given by (Equation 108). The expression in (Equation 113) represents the global operator in the form of a product of different quantum gates. The first part of the product in this expression is similar to the one obtained in [17]. The last part in the product is itself a product of the following gates:
(114)$$\begin{array}{cc} D_{1243}\left(ht\right)= & 1+\left(\cos(ht)-1\right)\left(|1100\rangle \langle 1100|+|0011\rangle \langle 0011|\right)\\ & -i\sin\left(ht\right)\left(|0011\rangle \langle 1100|+|1100\rangle \langle 0011|\right) \end{array}$$
(115)$$\begin{array}{cc} D_{1423}\left(ht\right)= & 1+\left(\cos(ht)-1\right)\left(|1001\rangle \langle 1001|+|0110\rangle \langle 0110|\right)\\ & -i\sin\left(ht\right)\left(|1001\rangle \langle 0110|+|0110\rangle \langle 1001|\right) \end{array}$$

The corresponding matrices of these two last unitaries are sparse matrices with four non-zero elements and can be easily implemented in practical quantum circuits. We want to emphasize that our method is direct because the Hamiltonians given in (Equation 111) and (112) as well as their exponential in (Equation 113) can be directly translated into quantum circuits without necessitating the intermediate stage employing the JWT (or the BKT). In our method, the fermionization obtained by the JWT is built in in the GA formalism.

## 5. GA Representation of the Quantum Ising and XY Hamiltonians

The quantum Ising Hamiltonian serves as a versatile and widely studied model in quantum simulation, providing valuable insights into the behavior of complex quantum systems and serving as a test bed for developing new simulation techniques and exploring emergent phenomena. While it may not capture all the complexities of real-world quantum systems, its simplicity, universality, and experimental feasibility make it an interesting tool for quantum simulation research [13,34,35].

Here we adopt a general approach, inspired from [34,35], considering a general anisotropic transverse *Z* field XY Hamiltonian, where the quantum Ising Hamiltonian with transverse *Z* field is a particular case (see hereafter). Both Hamiltonians include a term known as the transverse field, which introduces quantum mechanical effects. The general anisotropic transverse field XY Hamiltonian is given by:(116)$$\begin{array}{c} H=\frac{1}{2}\sum_{i=1}^{n-1}\left(\left(1+\gamma \right)X_{i}X_{i+1}+\left(1-\gamma \right)Y_{i}Y_{i+1}\right)+\lambda \sum_{i=1}^{n}Z_{i}. \end{array}$$

The first sum accounts for the degree of anisotropy interaction energy, and the second one for the transverse *Z* field. γ is the anisotropic coefficient, and taking γ=±1 leads to the transverse *Z* field Ising Hamiltonian, while taking γ=0 leads to the isotropic XX model. The λ factor is the strength of the transverse *Z* field. Using the reverse JWT, we can write this Hamiltonian in the GA formalism as:
(117)$$\begin{array}{cc} H & =\frac{1}{2}\sum_{i=1}^{n-1}\left(\left(1+\gamma \right)\left(c_{i}^{†}+c_{i}\right)\left(c_{i}c_{i}^{†}-c_{i}^{†}c_{i}\right)\left(c_{i+1}^{†}+c_{i+1}\right)-\left(1-\gamma \right)\left(c_{i}^{†}-c_{i}\right)\left(c_{i}c_{i}^{†}-c_{i}^{†}c_{i}\right)\left(c_{i+1}^{†}-c_{i+1}\right)\right)\\ & \;+\lambda \sum_{i=1}^{n}\left(c_{i}c_{i}^{†}-c_{i}^{†}c_{i}\right)\\ & =\frac{1}{2}\sum_{i=1}^{n-1}\left(\left(1+\gamma \right)\left(c_{i}^{†}-c_{i}\right)\left(c_{i+1}^{†}+c_{i+1}\right)-\left(1-\gamma \right)\left(c_{i}^{†}+c_{i}\right)\left(c_{i+1}^{†}-c_{i+1}\right)\right)+\lambda \sum_{i=1}^{n}\left(c_{i}c_{i}^{†}-c_{i}^{†}c_{i}\right)\\ & =\sum_{i=1}^{n-1}\left(\left(c_{i}^{†}c_{i+1}+c_{i+1}^{†}c_{i}\right)+\gamma \left(c_{i}^{†}c_{i+1}^{†}-c_{i}c_{i+1}\right)\right)+\lambda \sum_{i=1}^{n}\left(1-2c_{i}^{†}c_{i}\right)\\ & =\sum_{i=1}^{n-1}\left(c_{i}^{†}c_{i+1}+c_{i+1}^{†}c_{i}\right)+\gamma \sum_{i=1}^{n-1}\left(c_{i}^{†}c_{i+1}^{†}-c_{i}c_{i+1}\right)+n\lambda -2\lambda \sum_{i=1}^{n}c_{i}^{†}c_{i}. \end{array}$$

Then, by taking $e^{-iHt}$, the first sum gives the excitation operators. The nλ term reduces to a global phase shift, which is meaningless. The last sum gives the number operators.

In the second sum (including the γ factor), we recognize an excitation operator that undergoes the transformation $U\mapsto X_{i}UX_{i}$. Indeed, we have:$$\begin{array}{cc} X_{i}\left(c_{i}^{†}c_{i+1}^{†}-c_{i}c_{i+1}\right)X_{i} & =\prod_{j=0}^{i-1}Z_{j}\left(c_{i}^{†}+c_{i}\right)\left(c_{i}^{†}c_{i+1}^{†}-c_{i}c_{i+1}\right)\prod_{j=0}^{i-1}Z_{j}\left(c_{i}^{†}+c_{i}\right)\\ \; & =\left(c_{i}^{†}+c_{i}\right)\left(c_{i}^{†}c_{i+1}^{†}-c_{i}c_{i+1}\right)\left(c_{i}^{†}+c_{i}\right)\\ \; & =\left(c_{i}c_{i}^{†}c_{i+1}^{†}-c_{i}^{†}c_{i}c_{i+1}\right)\left(c_{i}^{†}+c_{i}\right)\\ \; & =c_{i+1}^{†}c_{i}+c_{i}^{†}c_{i+1}. \end{array}$$

Then, the quantum circuit can be written by using:(118)$$\begin{array}{c} e^{-iHt}≃{\left(e^{in\lambda }\prod_{i=1}^{n-1}e^{-iH_{ex_{i,i+1}}\Delta t}\prod_{i=1}^{n-1}X_{i}e^{-iH_{ex_{i,i+1}}\Delta t}X_{i}\prod_{i=1}^{n}T_{i}\left(-2\lambda \right)\right)}^{\frac{t}{\Delta t}}. \end{array}$$
where the $e^{-iH_{ex_{i,i+1}}\Delta t}$ has been calculated in (Equation 99).

The corresponding circuit diagram is given in Figure 6.

## 6. Conclusions

In this article, we have shown that Geometric Algebra (GA) using the Witt basis (WB) approach is equivalent to the Jordan–Wigner Transformation (JWT) commonly used in quantum simulation. This shows that the formalism of GA can be used to perform operations on Hamiltonians expressed in the second quantized form and to translate them into quantum circuits.

The important issue of our method is to propose a direct computational method in quantum simulation using GA compared to other methods where the direct representations of the excitation operators using quantum circuits cannot be systematically derived, requiring work on each particular case.

GA provides a clear and intuitive framework for representing quantum states and operations and allows a geometric interpretation of quantum operations. This geometric perspective can provide deeper insights into the behavior of quantum systems and facilitate the design and analysis of quantum algorithms and circuits. Dirac notation is translated to GA in a natural way: all expressions are represented in a particular algebra and thus may be manipulated and implemented as algebra elements directly, without any need for matrix representation as is sometimes necessary within the Dirac notation formalism.

Further directions could be explored to improve the method, for example, by further investigating the relation between the geometric product of GA and the tensor product between quantum states in the context of quantum computing. Concerning the Witt Basis (WB) approach, the question rises: is there a more general physical meaning and why does the WB structure describe the fermion second quantization representation so well?

The advantages of GA in scaling compared to qubit methods may vary depending on the specific application and context. GA can lead to more efficient representations of quantum states and operations compared to traditional approaches, such as matrix representations. This reduced complexity can help mitigate the challenges associated with scaling quantum algorithms to larger systems, including computational overhead and resource constraints. Research in this area is ongoing, and further developments may uncover additional advantages of GA in quantum computing scalability.

While JWT has been widely used in quantum computing qubit gate models for simulating fermionic systems, it can introduce stability challenges due to its complexity and error-prone nature. Our approach naturally accommodates the representation of fermionic states and operators and thus offers a more direct, natural, and potentially more stable alternative for representing fermionic systems in quantum computing applications. Also, our method could be applied for a GA representation of the Bravyi–Kitaev transformation, which is also widely used in quantum simulation because it seems less consuming in quantum gate resources.

We plan in the future to investigate some algorithmic implementation of the method presented here, for example, for the determination of molecule ground states.

## Figures and Tables

**Figure 1 entropy-26-00410-f001:**
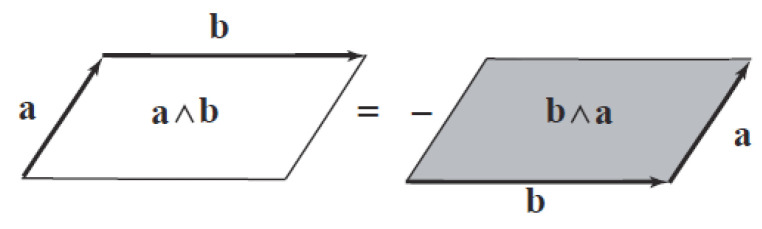
Representation of a bivector.

**Figure 2 entropy-26-00410-f002:**
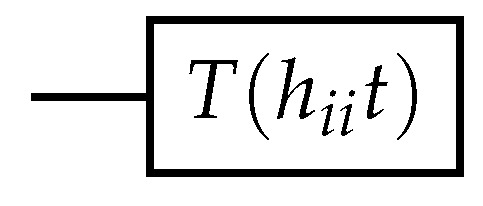
Quantum circuit for exponential of the number operator.

**Figure 3 entropy-26-00410-f003:**
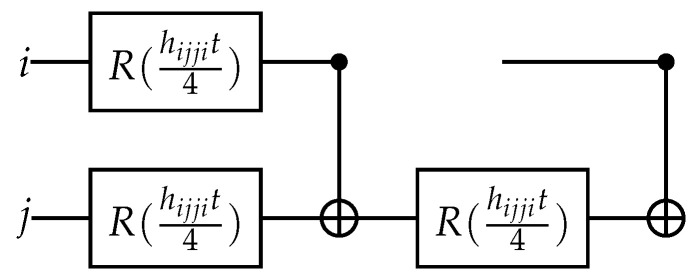
Quantum circuit for exponential of the Coulomb operator.

**Figure 4 entropy-26-00410-f004:**
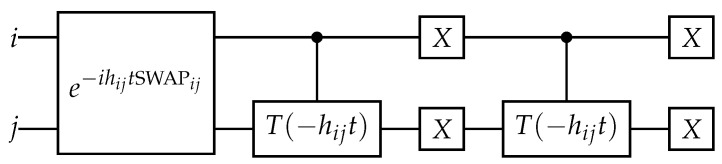
Quantum circuit for exponential of the Excitation operator assuming j=i+1.

**Figure 5 entropy-26-00410-f005:**
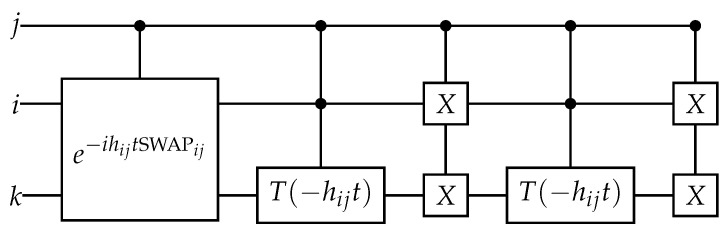
Quantum circuit for exponential of the excitation-number operator.

**Figure 6 entropy-26-00410-f006:**
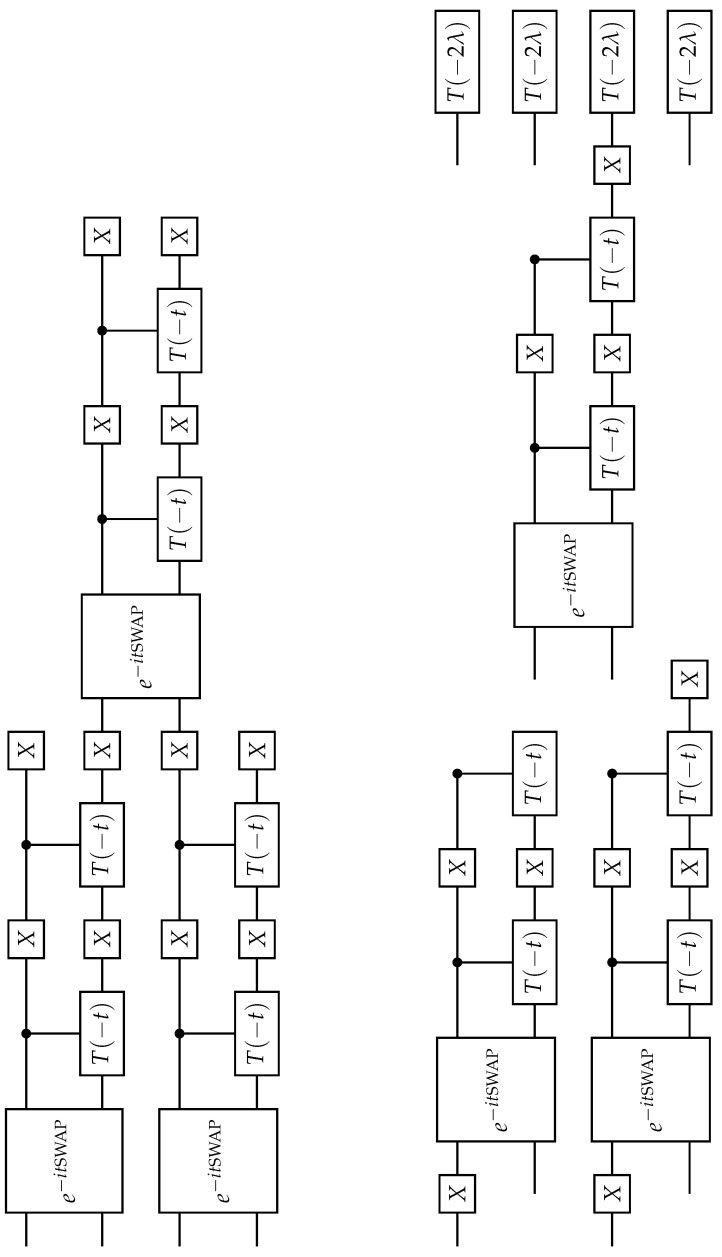
Quantum circuit for the quantum anisotropic Ising model simulation given in (Equation 117).

**Table 1 entropy-26-00410-t001:** Expression of the two-term product $\lambda_{j-1}|x_{j-1}\rangle$ for all possible factors. In the left-most column are the four possible left factors, $\lambda_{j-1}$, in the product, and in the upper-most line the two possible right factors, $|x_{j-1}\rangle$, in the product.

right factor	$c_{j-1}c_{j-1}^{†}$	$c_{j-1}^{†}$
left factor		
$c_{j-1}$	0	$\lambda_{j-1}c_{j-1}^{†}$
$c_{j-1}^{†}c_{j-1}$	0	$\lambda_{j-1}c_{j-1}^{†}$
$c_{j-1}^{†}$	$\lambda_{j-1}$	0
$c_{j-1}c_{j-1}^{†}$	$\lambda_{j-1}$	0

**Table 2 entropy-26-00410-t002:** Usual operators of a molecular Hamiltonian.

Operator	Second Quantized Form
Energy number operator	$H_{n_{i}}=h_{ii}c_{i}^{†}c_{i}$
Coulomb/exchange operator	$H_{c_{ij}}=h_{ijji}c_{i}^{†}c_{j}^{†}c_{j}c_{i}$
Excitation operator	$H_{ex_{ij}}=h_{ij}\left(c_{i}^{†}c_{j}+c_{j}^{†}c_{i}\right)$
Number-excitation operator	$H_{nex_{ijk}}=h_{ijjk}\left(c_{i}^{†}c_{j}^{†}c_{j}c_{k}+c_{k}^{†}c_{j}^{†}c_{j}c_{i}\right)$
Double excitation operator	$H_{dex_{ijk\ell }}=h_{ijk\ell }\left(c_{i}^{†}c_{j}^{†}c_{k}c_{\ell }+c_{\ell }^{†}c_{k}^{†}c_{j}c_{i}\right)$

## Data Availability

The data presented in this study are available on request from the corresponding author.

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
