# Peer review of "Geometric Algebra Jordan–Wigner Transformation for Quantum Simulation"

_entropy, 2024, doi:10.3390/e26050410_

Round 1

Reviewer 1 Report

Comments and Suggestions for Authors

In this paper, the authors proposed to a scheme to the investigation of a quantum simulation circuit of utilizing the Witt basis method in Geometric Algebra to reformulate the Jordan-Winger transformation to express various quantum gates. Meanwhile, they rewrite the general one and two-electron Hamiltonian and propose a design of a quantum simulation circuit for Hydrogen molecule. The paper is interesting and reasonable. I suggest that this paper be published on Multidisciplinary Digital Publishing Institute, but the following issues need to be dealt with before that:

1. The term arising in the first row in the table 1 has a format error, this should be correct.

2. Operators in the article need to add operator symbols for easy recognition.

3. In line 113, “the hamiltonians given in 113 and 114” in which the number 113 and 114 should be convert Eq. (113) and Eq. (114).

4. The authors shown that GA using the Witt basis approach is equivalent to the JWT. Why we choose the GA for quantum simulation?

5. What is the fidelity for the quantum simulation by GA?

6. Several two-qubit gates are applied for quantum simulation. Is it possible to realize it in experiment?

Comments on the Quality of English Language

OK

Author Response

Reply to comments and suggestions of reviewers

We would like to thank the reviewers for their valuable comments and suggestions and for their time.

We revised the paper following the remarks and suggestions, the corresponding parts are highlighted in blue in the revised version.

Following comments and suggestions of reviewer 1:

We clarified the expressions in table 1 by better specifying the different role of the terms in the figure caption.

We gave in the conclusion the reasons why we think that GA brings a benefit over traditional JWT applications in quantum simulation.

We want to clarify the fact that in GA operators and vectors are treated at the same level so we think that there is no need to distinguish operators using a special notation.

We corrected notations for equations.

In future work we emphasize to realize algorithmic implementations permitting to calculate explicitly quantum circuit fidelity taking into account error models and assumed imperfections and we will compare our circuit to real experimental realizations. Here in this paper we are at the model level so we did not analyze fidelity.

Best Regards

The Authors

Reviewer 2 Report

Comments and Suggestions for Authors

In this submitted manuscript, the authors present a way to reformulate Jordan-Wigner transform into geometric algebra, use this formulation to represent a few quantum states and then apply this formulation to one- and two-electron integral of a hydrogen molecule. While introducing geometric algebra looks novel in this field, I feel there is a main lack of motivation. There also exist other issues. I would reconsider this manuscript if the authors can address my questions below.

Why do we need GA and why is this an important introduction? Would there be an improvement on number of qubits or scaling? Or is this formulation more stable than the typical JW or BK formulations in the literature?

The authors present an example using hydrogen molecule. This is a super simple example as hydrogen molecule has been in quantum chemistry textbook for many decades. H2 can even been studied with your pen on papers. The author shows that GA can be used for H2, which is indeed a molecule but this is far from the typical bottleneck/true molecular problems people are trying to solve. Can there be another example, at least more difficult than H2?

Many problems in the introduction part

1.         The major goal in computational chemistry is to find the ground-state energies of many body interacting fermionic systems.”

This statement is too absolute. First, ground state energy itself is meaningless, only the differences between ground state energy and excited state energies (spectrum) are meaningful and represents the spectral properties of the system. Second, while the spectrum is useful, the wavefunctions and structural properties are also important.

2.         tensor networks do not provide a great improvement when the system is too much entangled

What is the better method than tensor network when the system is too much entangled, in your opinion?

3.         They can provide an exponential advantage over classical computers”

What is the exponential advantage? Advantage in memory? Advantage in scaling? In my mind, quantum computers are promising. However, quantum chemistry problems are in QMA class and there has not been any proof about the scaling.

Comments on the Quality of English Language

English quality is ok.

Author Response

Reply to comments and suggestions of reviewers

We would like to thank the reviewers for their valuable comments and suggestions and for their time.

We revised the paper following the remarks and suggestions, the corresponding parts are highlighted in blue in the revised version.

Following comments and suggestions of reviewer 2:

Regarding the 3 questions:

1) Why do we need GA and why is this an important introduction?

2)Would there be an improvement on number of qubits or scaling?

3) Or is this formulation more stable than the typical JW or BK formulations in the literature?

We added 3 new paragraphs in the conclusion part (highlighted in blue) discussing these topics.

Regarding the remarks on the presentation of the H2 molecule model we added an entire new section (section 5) on a general model for the quantum Ising Hamiltonian which is largely used as a surrogate in quantum simulation problems. We developed GA expressions in equations (118-120) and proposed the corresponding quantum circuit (figure 6)

We also followed the suggestions and corrected and modified the discussions on quantum simulation in the introductory part.

Best Regards

The Authors

Round 2

Reviewer 2 Report

Comments and Suggestions for Authors

Thank the authors for the revision. I can see their efforts. However, I still see some issues listed below. Please revise and the newly revised version would look good to me.

In the first paragraph ‘direct simulations are only possible for molecules with few atoms’. This is definitely wrong. Density functional theory/MD etc have enabled calculations for systems > 10k atoms.

in order to ease the computations for complex molecules, such as quantum Monte Carlo [1], tensor networks 24

[2] or density functional theory’. Besides these, better mention quantum chemistry methods such as post Hartree Fock, Green’s function methods such as many-body perturbation theory etc.

Quantum computers are today a major research subject, involving both private companies and universities, all

around the world.’ Please select to cite paper/papers from e.g. IBM/Google

Adding a paragraph about quantum Ising chain is good. Question about this sentence: “For instance, the transverse field Ising model is a particular case of the anisotropic transverse-field XY model, whose Hamiltonian is”

It looks like you’re mostly discussing transverse XY model in this paragraph. Why mentioning transverse field Ising at the beginning?

The new sentences in conclusion part are good.

Comments on the Quality of English Language

Some sentences have grammar issue/typos. Please check and revise.

Author Response

We would like to thank the reviewer for her/his valuable comments and suggestions and for her/his time in this second round revision.

We revised the paper following the remarks and suggestions:

We added the remarks on the different molecular simulation methods and added corresponding references.

We added references of papers on quantum simulation from the companies IBM, Google, Quantinuum and Pasqal

We clarified in section 5 the way we introduce a general form of the anisotropic transverse XY Hamiltonian which as a particular case, for the coefficient γ=±1, gives the quantum Ising Hamiltonian.

Best Regards

The Authors